# The Physiological Role of Abscisic Acid in Regulating Root System Architecture of Alfalfa in Its Adaptation to Water Deficit

**Shuo Li [1,2], Zhongnan Nie [3], Juan Sun [1], Xianglin Li [2,\*] and Guofeng Yang [1,\*]**

[1]  Key Laboratory of National Forestry and Grassland Administration on Grassland Resources and Ecology in the Yellow River Delta, College of Grassland Science, Qingdao Agricultural University, Qingdao 266109, China

[2]  Institute of Animal Science, Chinese Academy of Agricultural Sciences, Beijing 100193, China

[3]  Department of Jobs, Precincts and Regions, Private Bag 105, Hamilton, VIC 3300, Australia

[\*]  Correspondence: author: lxl@caas.cn (X.L.); yanggf@qau.edu.cn (G.Y.)

**Abstract:** Alfalfa (*Medicago sativa* L.) is a perennial leguminous plant, with a strong tap root system that plays an important role in alfalfa's adaptation to drought stress. However, a better understanding of root functional traits and how these root traits are related to whole plant responses in order to improve pasture productivity under water deficit. Two greenhouse experiments were conducted: Experiment 1 used three alfalfa cultivars and four levels of soil water content treatments to investigate herbage productivity, growth point density, residual shoot weight, and root weight. Experiment 2 assessed relationships among root-sourced abscisic acid (ABA), root system architecture and plant biomass in response to water deficit. The results demonstrated that root system was used as a useful tool to improve tolerant and adaptation when alfalfa copied with lower levels of soil water content. On average, maintaining 60–65% soil water-holding capacity alfalfa had the highest herbage accumulation (6.7 g DM pot-1), growing point density (46.5 pot-1), and residual shoot biomass (1.8 g DM pot-1). At the level of water stress, *Medicago sativa* L. cv Zhaodong (ZD) and cv Aohan (AH) tended to exhibit a herringbone branching pattern with less root tips, root forks, altitude, and magnitude than cv Golden empress (GE). Principal component analysis and structural equation modeling revealed that root-sourced ABA positively regulated the altitude and magnitude of root system architecture, root tips and root forks, and was closely associated with plant root biomass and herbage biomass. It was concluded that these findings can contribute to developing optimum irrigation strategies and help alfalfa breeders in the development of new cultivars with improved drought tolerance based on root system architecture, plant hormone, and plant growth.

**Keywords:** water deficit; herbage accumulation; root abscisic acid; root system architecture

## 1. Introduction

Alfalfa (*Medicago sativa* L.) is an important perennial forage legume with high nutritional quality and high average yields. It has been widely grown in Northwest China for many years [1–3]. The planting area of alfalfa in the region accounts for approximately 80% of the total cultivated grassland, making an important contribution to the local development of agriculture and animal husbandry as well as ecosystem services [4]. A deep understanding of the crucial drought-adaptive mechanisms of alfalfa root system should be of substantial agricultural importance in Northwest China.

Global climate change models indicate that water shortages are responsible for the greatest crop losses worldwide, heightening international interest in crop drought tolerance [5,6]. Limitation of water availability is a critical factor in agricultural production [7] and is predicted to worsen under future climate change [8], especially in semi-arid and

arid regions [9]. Drought stress consistently results in a decline in yields and productivity [10,11]. The current trend for droughts in China is superimposed with human-induced global warming, which is predicted to enhance aridity in Northwest China [12]. The increasing incidence of drought in the northwestern region of China has reduced the total area of alfalfa planting and poses a serious threat to alfalfa germplasm resources and forage productivity [3]. More research is needed to fully understand the persistence traits of alfalfa both in semi-arid and arid environments and environments with water shortage.

Plant growth, development, and yield performance differ among crops subjected to drought stress under field conditions [13,14]. These differences in response to abiotic stresses may be closely associated with root morphological traits and root functions [15]. Roots, as one of the primary organs, play an important role in vascular plants. They are a hidden part of a plant, responsible for growth, development and productivity, anchorage, and supplying stem and leaves with water and nutrients [16,17]. Acquisition of water (and mineral nutrients) is primarily determined by the dimension of the entire root zone, root density, their differentiation and elongation, and is closely related to soil water availability [18,19]. Roots exhibit a high degree of morphological plasticity in response to soil physical properties. Root topology and fractal geometry have recently attracted increasing research interest [20]. The topological structure is an important component of root system architecture, determines the spatial distribution of root systems in the soil, and affects the capacity for water uptake and nutrient absorption [21,22]. Li et al. [3] proposed that root topology comprises two extreme branching patterns (dichotomous and herringbone branching) under abiotic stress conditions. The fractal dimension (FD) is correlated with root topology [3,20] and root system architecture [23]. Root tips may be more important for the uptake of mobile resources. Characterization of the morphology and root systems architecture assist in understanding the functional differences and growth strategy of root systems among species as well as in screening for genetic improvement of crops [24]. At present, the study of plant root responses to water stress has focused on the root morphology, and a quantitative description of root topology structure and fractal characteristics is lacking. Thus, an area of recent interest is improvements of root traits that increase efficient deployment of tissues for foraging of soil water and, expressly, the maintenance of productivity under water deficit.

Root growth and development processes are regulated by a number of phytohormones, including auxin, cytokinins (CTKs), abscisic acid (ABA), gibberellins (GAs), and ethylene (ETH) [25–28]. Hormones associated with stress responses also regulate growth of the plant. In recent years, ABA has been shown to play important roles in the regulation of root growth and development [29–34]. ABA is an essential hormone to maintain root growth in both well-watered and drying soil [35]. ABA has a much stronger effect on lateral root than primary root growth, suggesting that differences in environmental responsiveness between these two root types may be due to divergent hormone signaling networks [29,32,35]. However, there has been little integrated assessment of how root ABA regulates root system growth and development in response to drought stress. The objectives of this study were (1) to assess the response of alfalfa to different water stress treatments, and the effects on herbage accumulation, growth point density, residual shoot, and root weight in three drought-resistant roots under long-term soil drought; (2) to evaluate the direct and indirect roles of root-sourced ABA in regulating root system architecture and plant biomass; and (3) to understand how water deficit modulates root system architecture, root spatial distribution, and plant hormone, which help to understand plant drought resistance.

## 2. Materials and Methods

### 2.1. Experimental Site and Materials

Two experiments were conducted at the Institute of Animal Science, Chinese Academy of Agricultural Sciences, Beijing, China. Experiment 1 was a pot experiment to

measure aboveground and belowground biomass and growing point density under water deficit from January to June 2019. Experiment 2 was conducted from September to December 2019 to investigate root system architecture, root morphological traits, plant biomass, and root ABA content under water stress. All plants were grown in a greenhouse with controlled temperature and light intensity (25-30 °C and 118-145 W/m²).

The alfalfa cultivars used for the experiments were *Medicago sativa* cv. 'Aohan' (AH), 'Golden Empress' (GE), and 'Zhaodong' (ZD), which are all commonly grown in Northwest China. ZD grows well in the semi-arid areas with annual rainfall of 350 to 800 mm, annual evaporation of 2534 mm, and annual average temperature of 6.4 °C. GE is an indigenous forage grown in the semi-arid regions with annual rainfall of 400 to 600 mm, annual evaporation of 1500 mm, and annual average temperature of 8.7 °C, and AH has been bred for high yield in the region [3]. Prior to the experiment, seeds of each cultivar were surface-sterilized with 1% NaClO for 30 min, then washed with deionized water four times, and placed in petri dishes at 50 seeds/petri dish [3,36]. Germinated seeds were transferred to an incubator at 25 °C for 48 h to allow the root to develop before being transplanted for Experiments 1 and 2.

### 2.2. Experiment 1

#### 2.2.1. Experimental Description and Design

On 7 January 2019, the seedlings were transplanted into plastic pots (18 cm diameter, 16 cm deep) and grown in a greenhouse under ambient temperature and natural light from January to June 2019. Each pot was filled with 0.7 kg potting mix and five equally spaced seedlings were planted per pot. The sterilized potting mix comprised a mixture of nutrient-enriched soil and perlite (2:1, *v/v*) with 91.3% organic matter and 6.4 pH. The pots were checked regularly and watered three times weekly to ensure the availability of sufficient moisture for growth until 22 February 2019. Weeds were removed manually and diseases were chemically controlled to ensure the seedlings remained healthy and vigorous prior to experimental treatment. The three cultivars and four levels of water management treatments were combined to form 3 × 4 = 12 factorial treatments, which were replicated four times in a randomized complete block design (RCBD). The water stress treatments comprised maintenance of the soil water content at (1) 75–80% of the water-holding capacity, (2) 60–65% of the water-holding capacity, (3) 45–50% of the water-holding capacity, and (4) 30–35% of the water-holding capacity. From 23 February 2019 onwards, all pots were weighed and adjusted to the specified water-holding capacity every two days. Four harvests were performed over the course of the experiment.

#### 2.2.2. Soil Water-Holding Capacity

At the beginning of the experiment, individual pots were weighed when the potting mix was dry. The pots were then watered to full saturation, drained freely for 48 h and weighed again. Soil water-holding capacity (gravimetric soil moisture) for each pot was determined by the difference between these two time weights and the weight of potting mix [3].

#### 2.2.3. Measurements

##### Herbage Accumulation

Herbage was harvested on four dates during the experimental period (22 March, 19 April, 20 May, and 18 June 2019). The herbage sample from each pot was oven dried at 65 °C until constant weight was attained [36,37]. Herbage mass was calculated as gram dry matter (DM) per pot, and the herbage mass at each harvest was summed to determine the herbage accumulation throughout the experiment.

Growing Point Density

At the end of Experiment 1, prior to the final herbage mass harvest, growing point density was determined by counting the total number of branches of tested plants per pot. Growing point density was recorded as number of growing points per pot [36].

Root and Residual Shoot Biomass

At the end of Experiment 1, all residual shoots were cut to the soil surface, then oven dried at 100 °C for 48 h and weighed. All root systems from individual pots were dug by hand, washed under running water to remove soil particles, then oven dried at 100 °C until constant weight was attained [38].

*2.3. Experiment 2*

2.3.1. Experimental Description and Design

Seedlings were transplanted into cylindrical pots (18 cm diameter, 50 cm deep) and grown in a greenhouse under ambient temperature and natural light from September to December 2019. A nylon mesh bag was placed in each pot, filled with 3.0 kg potting mix (sieved with 5 mm mesh, 6.0 pH), into which one seedling was transplanted. The sterilized potting mix contained a mixture of nutrient-enriched soil and sand (5:1, *v/v*) with 91.3% organic matter, 1.6% total nitrogen, 0.1% total $P_2O_5$, and 0.2% total $K_2O$. The pots were checked regularly and watered three times weekly to ensure sufficient moisture for growth until the seedlings were two months old. Experiment 2 used the same alfalfa cultivars, water level treatments and experimental design as Experiment 1, which were replicated ten times in a randomized complete block design (RCBD). Half of the replicated alfalfa seedlings were used to measure root-sourced ABA content, and the other half to examine the root system architecture.

2.3.2. Root Sampling and Scanning

At the end of the experiment, the nylon mesh bag containing plant roots was carefully removed from each pot to maintain the original spatial distribution and minimize damage and disturbance to the root system. The roots were washed thoroughly by hand, rinsed with deionized water, and then spread in 1 cm water in a clear acrylic tray and scanned at 600 dpi resolution for further analysis with a ScanMaker i800 Plus graphic scanner (Microtek, Hsinchu, Taiwan). Win-RHIZO 2017a software (Regent Instruments, Inc., Quebec, QC, Canada) was used to analyze the root system images and measure altitude, magnitude, number of root forks, and number of root tips. After scanning, the root dry weight was determined after the samples were oven dried at 65 °C for at least 48 h. A subset of roots washed with distilled water and dried with filter paper were frozen in liquid nitrogen and then stored at −80 °C for ABA content analysis.

2.3.3. Measurements

Root-Sourced ABA Content

The ABA content in the roots was quantified using a high-performance liquid chromatography-mass spectrometry system (HPLC-MS) and a Waters HSS T3 column (2.1 × 50 mm; 1.8 mm). The mobile phase A and B solvents consisted of methanol/0.1% methanoic acid and ultrapure water/0.1% methanoic acid, respectively. The injection volume was 2 μL, and column temperature was 40 °C. The MS conditions were as follows: spray voltage, 2800 V; pressures of the air curtain, nebulizer, and auxiliary gas were 15, 65, and 70 psi, respectively; and atomizing temperature was 400 °C [39].

Topological Index

The topological index (TI) is calculated to reflect the branching patterns of different plant roots and is defined as log altitude (A)/log magnitude (M) [3,40], where A is the total

number of internal links and M the total number of external links in the root system. Two extreme patterns of root topology were proposed: herringbone and dichotomous branching (Figure 1).

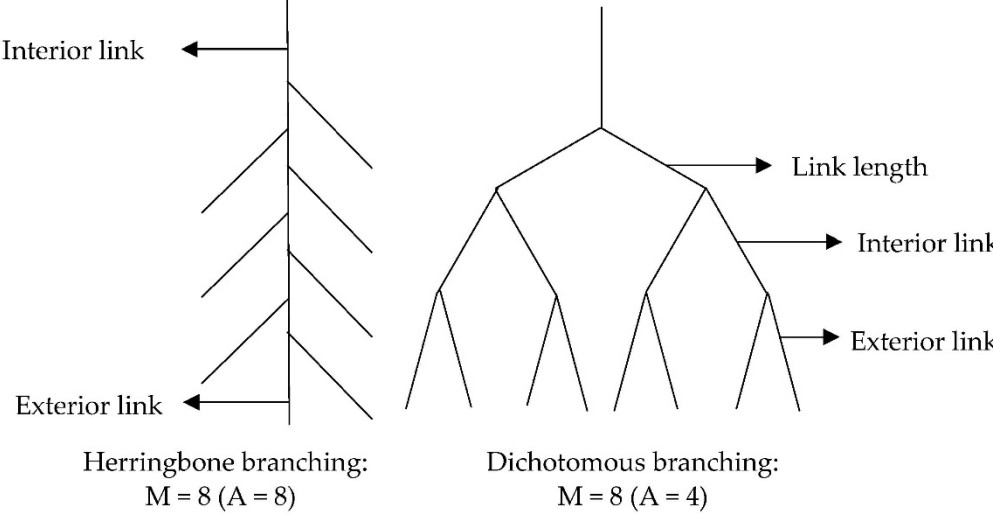

**Figure 1.** Schematic representation of Fitter's link-based parameters used to describe root topology. A link is defined as a piece of the root between two branching points (interior link) or between a branch and a meristem (exterior link). The magnitude (M) of the overall root system represents the number of exterior links, which equals the number of meristems in a root. The magnitude of an individual link within the root system represents the total number of root segments connected to the shoot through that specific link. The altitude of the overall root system (A) is the number of links in the longest path from an exterior link to the most basal link of the root system.

Fractal Analysis

Digitized root images were used for fractal analysis following the box-counting method [41] using Win-RHIZO 2017a software. Each root image was first covered with a frame that was divided into boxes (grids) with side length $r$. The size of the boxes was designated from 200 to 3200 pixels (0.008–0.127 mm). The number $N(r)$ of boxes that intersected with the image was counted. The number $N(r)$ of boxes gradually increased with decreasing side length $r$. When plotting $N(r)$ against $r$ on a lg–lg scale, the power–law relationship $N(r) = Kr - D$ was obtained if the image was fractal. The two constants $D$ and lg $K$ were calculated based on the equation lg $Nr = -FD$ lg$r$ + lg$K$ as the fractal parameters. The negative value of the slope of the regression line is the fractal dimension (FD) [3,24,42].

*2.4. Statistical Analyses*

The statistical significance of differences between means was analyzed for all parameters using the General Analysis of Variance (ANOVA) procedure of GENSTAT Release 12.0 (VSN International 2009). A RCBD model with repeated measures was used to analyze their interactions between cultivars and drought severity on herbage accumulation, growing point density, root weight, residual shoot weight, root ABA content, root system architecture, and plant biomass for Experiment 1 and 2 [36]. Structural Equation Modeling (SEM) was performed using AMOS 22.0 (Amos Development, Spring House, PA, USA) to analyze hypothetical pathways of direct and indirect regulation by root ABA on plant root traits and plant biomass under water deficit. All raw data (with water stress as the main effect) in the SEM included root ABA content, root morphological traits, TI, FD, root biomass, and herbage biomass. Data were fitted to the models using the maximum likelihood estimation method. The suitability of each model was evaluated using the chi-square ($\chi^2$) test, degrees of freedom (df), goodness-of-fit (GFI), and root mean square error of approximation (RMSEA). Significance was determined at $p < 0.05$ or 0.01 significant levels.

## 3. Results

### 3.1. Experiment 1

#### 3.1.1. Herbage Accumulation

Overall significant interactions and differences in alfalfa herbage accumulation were detected among soil water content treatments and alfalfa cultivars (Figure S1).

There was a significant interaction ($p < 0.01$) between soil water content and cultivars on herbage accumulation (Figure 2). The herbage accumulation of ZD was 6.6 g DM pot$^{-1}$ when the soil water content was at 60-65% of the water-holding capacity, which was 1.4-fold higher than that (5.2 g DM pot$^{-1}$) observed when the soil water content was at 75-80% of the water-holding capacity. The herbage accumulation of AH was 6.4 g DM pot$^{-1}$ when the soil water content was at 75–80% of the water-holding capacity, peaked at 7.2 g DM pot$^{-1}$ when the soil water content was at 60–65% of the water-holding capacity, and was 6.0 and 4.4 g DM pot$^{-1}$ under 45–50% and 30–35% of the water-holding capacity, respectively. However, the herbage accumulation of GE generally declined with decreasing soil water content.

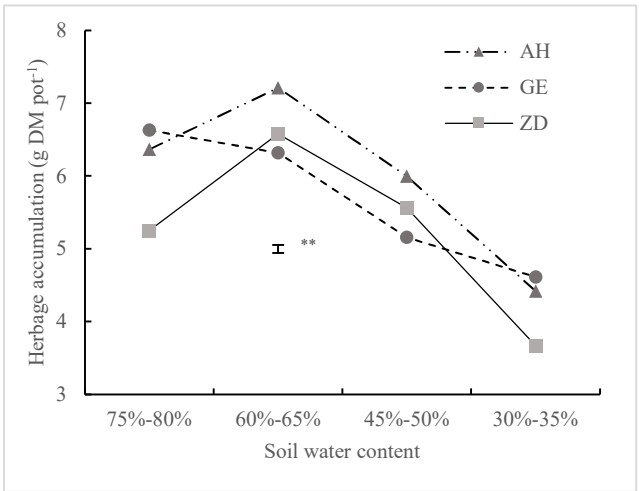

**Figure 2.** Interaction of soil water content (75–80%, 60–65%, 45–50%, and 30–35% water-holding capacity) and cultivars (AH: Aohan; GE: Golden Empress; ZD: Zhaodong) on herbage accumulation of alfalfa. The error bar on the figure represents standard error of the mean, ** $p < 0.01$.

#### 3.1.2. Growing Point Density

There were significant ($p < 0.01$) differences in growing point density of alfalfa among soil water content treatments and cultivars (Figure S2).

There was a significant ($p < 0.01$) interaction on growing point density between soil water content and cultivars (Figure 3). Growing point density of all cultivars was highest (mean = 46.5 growing points pot$^{-1}$) when soil water content was at 60–65% of the water-holding capacity. Growing point density of GE and ZD increased sharply, whereas that of AH increased slightly, when soil water content changed from 75–80% to 60–65% of the water-holding capacity. The growing point density of all cultivars declined when soil water content decreased from 60–65% to 30–35% of the water-holding capacity.

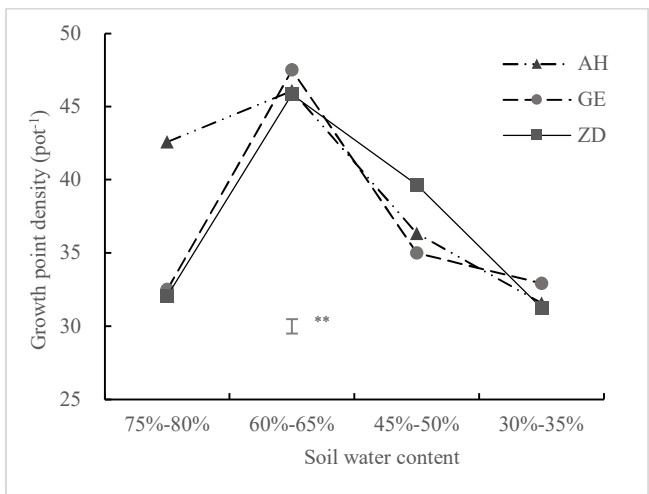

**Figure 3.** Interaction between soil water content (75–80%, 60–65%, 45–50%, and 30–35% of water-holding capacity) and cultivars (AH: Aohan; GE: Golden Empress; ZD: Zhaodong) on alfalfa growing point density (GD). ** $p < 0.01$.

### 3.1.3. Residual Shoot Biomass and Root Biomass

There were significant ($p < 0.01$) differences in residual shoot and root biomass among soil water content and cultivars (Figure S3).

There were significant ($p < 0.01$) interactions on residual shoot and root biomass between soil water content and cultivars (Figure 4a,b). The residual shoot biomass of all cultivars increased initially from 75–80% to 60–65% of the water-holding capacity and declined thereafter with decreasing soil water content; however, the magnitude of decline for ZD was much greater than that for AH and GE (Figure 4a). Under 60–65% of the water-holding capacity, the residual shoot biomass peaked (mean = 1.8 g DM pot$^{-1}$), which was on average 20% higher than that under 75–80% and 45–50% of the water-holding capacity, and 50% higher than that under 30–35% of the water-holding capacity. The root biomass was highest for ZD and GE (1.4 and 1.3 g DM pot$^{-1}$, respectively) when soil water content was at 60–65% of the water-holding capacity, whereas the root biomass of AH was highest (1.6 g DM pot$^{-1}$) at 75–80% of the water-holding capacity (Figure 4b).

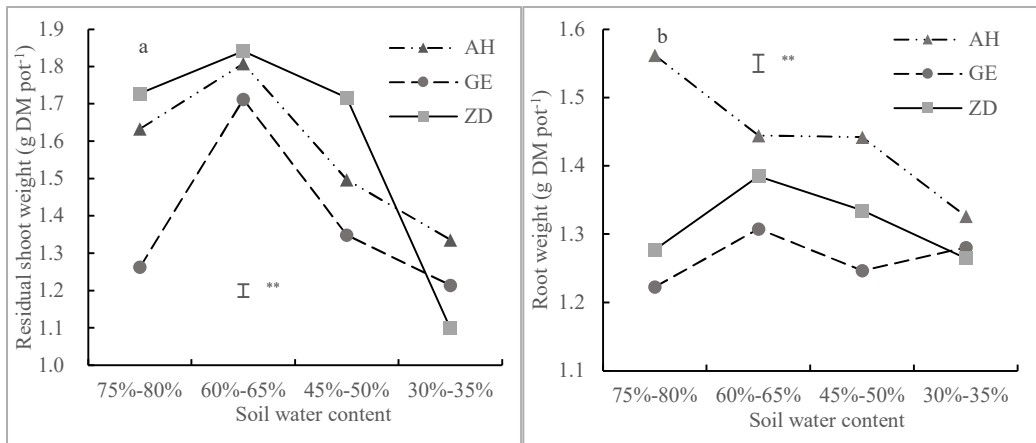

**Figure 4.** Interaction between soil water content (75–80%, 60–65%, 45–50%, and 30–35% of water-holding capacity) and cultivars (AH: Aohan; GE: Golden Empress; ZD: Zhaodong) on (**a**) residual shoot weight and (**b**) root weight of alfalfa. ** $p < 0.01$.

### 3.2. *Experiment 2*

#### 3.2.1. Root-Sourced ABA Content

There was a significant interaction on root ABA content between cultivar and soil water content (Figure 5). The content of ABA was peaked for three alfalfa cultivars when soil water content was at 30–35% of the water-holding capacity. In contrast, root-sourced ABA content was less for all cultivars under 75–80% of the water-holding capacity. Root-sourced ABA content generally increased with decreasing soil water content for all cultivars. However, the increase was almost linear for GE, but curved for AH and ZD.

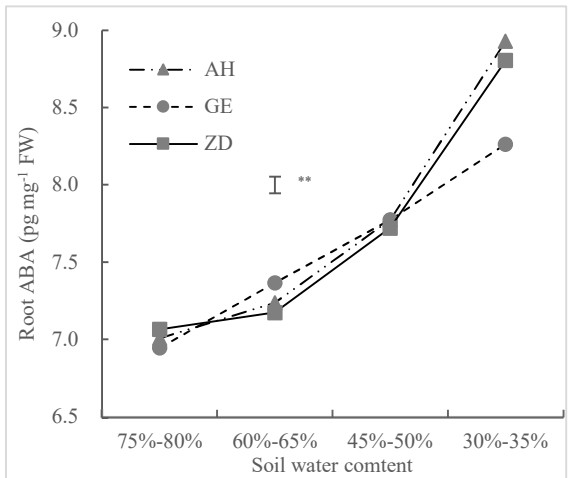

**Figure 5.** Interaction between soil water content (75–80%, 60–65%, 45–50%, and 30–35% of water-holding capacity) and cultivars (AH: Aohan; GE: Golden Empress; ZD: Zhaodong) on root abscisic acid (ABA) content of alfalfa. ** $p < 0.01$.

#### 3.2.2. Root Morphology, Topology, and Root Fractal Analysis

There were significant ($p < 0.01$) interactions on the numbers of root forks and root tips between soil water content and cultivars (Figure 6a,b). A consistent upward trend in the number of root forks was observed for GE with decreasing soil water content. However, the number of root forks and root tips for AH and ZD decreased initially and then increased with declining soil water content from 75–80% to 30–35% of the water-holding capacity (Figure 6a). An upward trend in the number of root tips for GE was observed with decreasing soil water content (Figure 6b). However, similar to root forks, the number of root tips for AH and ZD decreased initially and then increased with declining soil water content from 75–80% to 30–35% of the water-holding capacity (Figure 6b).

Cultivars and soil water content had significant ($p < 0.01$) interactions on altitude and magnitude (Figure 6c,d). The attitude of GE increased with declining soil water content whereas that of AH and ZD declined initially and then increased when soil water content decreased from 75–80% to 30–35% of the water-holding capacity (Figure 6c). The magnitude of cultivars showed a similar trend (Figure 6d).

There was a significant ($p < 0.01$) interaction on TI between soil water content and cultivars (Figure 6e). There was a downward trend of TI with decreasing soil water content for GE. However, the TI of AH and ZD peaked at 60–65% of the water-holding capacity (mean = 0.62), and was reduced on average by 6.5%, 8.1%, and 8.1% at 75–80%, 45–50%, and 30–35% of the water-holding capacity, respectively (Figure 6e).

There was a significant ($p < 0.01$) interaction on FD between soil water content and cultivars (Figure 6f). The FD of GE increased with decreasing soil water content whereas that of AH and ZD declined initially and then increased when soil water content declined from 75–80% to 30–35% of the water-holding capacity (Figure 6f).

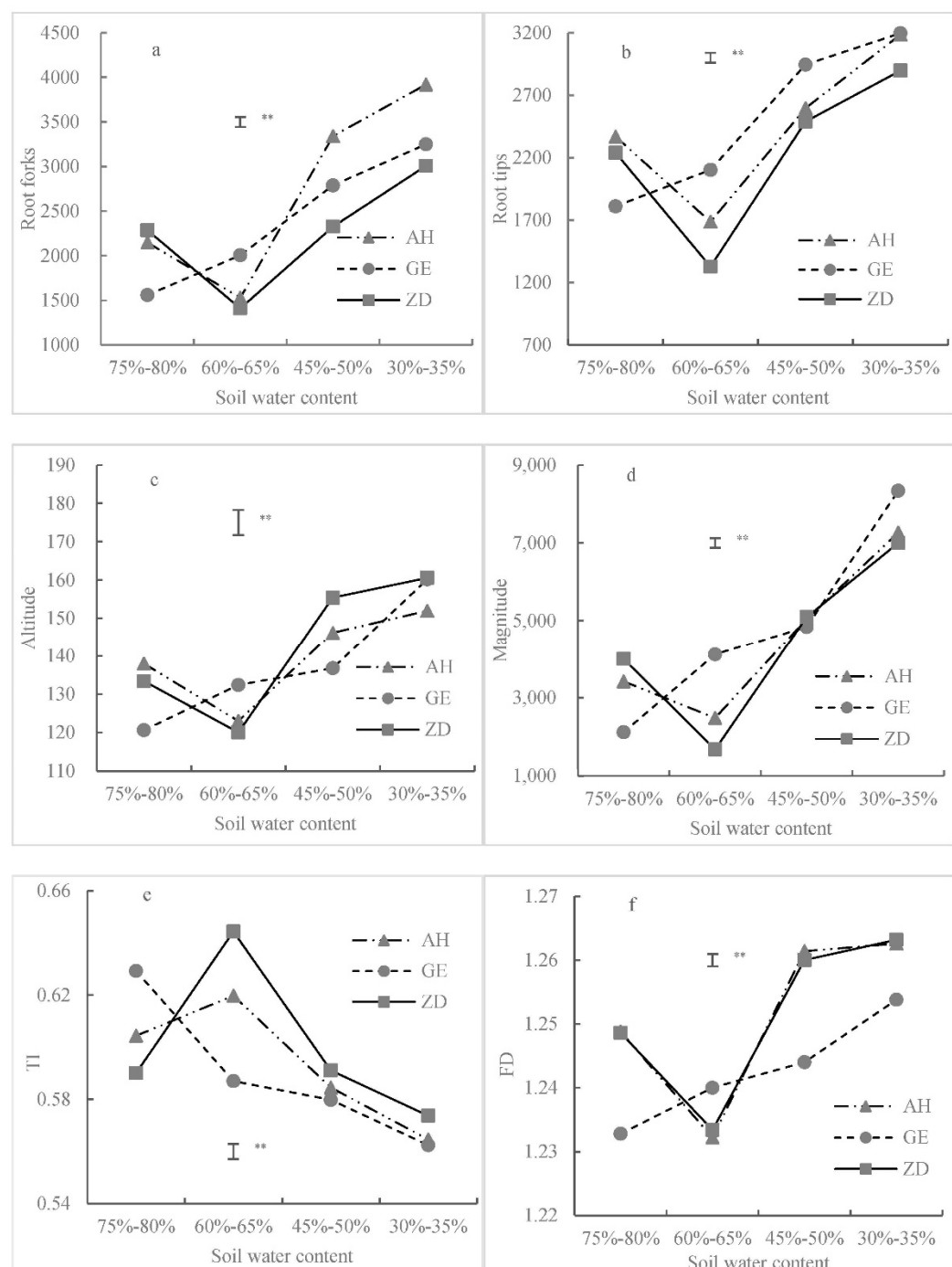

**Figure 6.** Interactions between soil water content (75–80%, 60–65%, 45–50%, and 30–35% of water-holding capacity) and cultivars (AH: Aohan; GE: Golden Empress; ZD: Zhaodong) on (**a**) number of root forks, (**b**) number of root tips, (**c**) altitude, (**d**) magnitude, (**e**) topological index (TI), and (**f**) fractal dimension (FD) of the root system of alfalfa. ** $p < 0.01$.

### 3.2.3. Root Biomass and Herbage Biomass

There were significant ($p < 0.05$ or $0.01$) interactions on root and herbage biomass between cultivars and soil water content (Figure 7a,b). In general, increase in root biomass of GE was associated with decreasing soil water content. Under 60–65% of the water-holding capacity, AH and ZD showed the lowest root biomass (mean 0.2 g DM pot$^{-1}$). The root biomass of all cultivars peaked (mean 0.6 g DM pot$^{-1}$) under 30–35% of the water-holding capacity (Figure 7a).

There was a trend of decline in herbage biomass of GE with decreasing soil water content. The herbage biomass of AH and ZD increased initially and declined thereafter with declining soil water content from 75–80% to 30–35% of the water-holding capacity. The herbage biomass of AH and ZD peaked (mean 4.0 g DM pot$^{-1}$) under 60–65% of water-holding capacity, and was reduced by 12.5%, 8.8%, and 16.3% under 75–80%, 45–50%, and 30–35% of the water-holding capacity, respectively (Figure 7b).

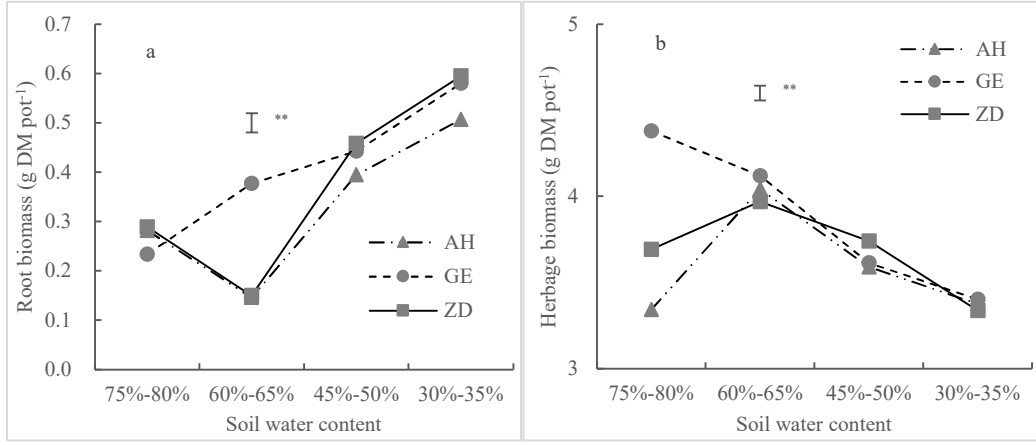

**Figure 7.** Interactions between soil water content (75–80%, 60–65%, 45–50%, and 30–35% of water-holding capacity) and cultivars (AH: Aohan; GE: Golden Empress; ZD: Zhaodong) on (**a**) root biomass and (**b**) herbage biomass of alfalfa. ** $p < 0.01$.

### 3.2.4. Modelling Analysis of Root ABA Content, Root System Architecture, and Plant Biomass

The PCA ordination diagram showed that root system characteristics differed among various soil water content treatments. The data points for each soil water content treatment were clustered, especially for the 30–35% of the water-holding capacity (Figure 8). Root ABA content was positively associated with altitude, magnitude, number of root tips and forks, FD, and root biomass, and was negatively associated with herbage biomass and TI. TI and herbage biomass were negatively associated with altitude, magnitude, root forks, root tips, FD, and root biomass. The PC1 and PC2 axes explained 97.3% of the total variation under the water deficit treatments (Figure 8).

Based on the PCA analysis, root ABA content was significantly ($p < 0.01$) positively correlated with altitude, magnitude, and number of root forks and tips under various soil water content treatments (Figure 9). The FD showed significant ($p < 0.05$) positive correlations with the number of forks, number of root tips and root biomass, and a significant ($p < 0.05$) negative correlation with herbage biomass, under water stress. Altitude showed a significant ($p < 0.01$) positive correlation with TI, and magnitude was negatively ($p < 0.01$) correlated with TI. The TI was also negatively ($p < 0.01$) correlated with root biomass (Figure 9).

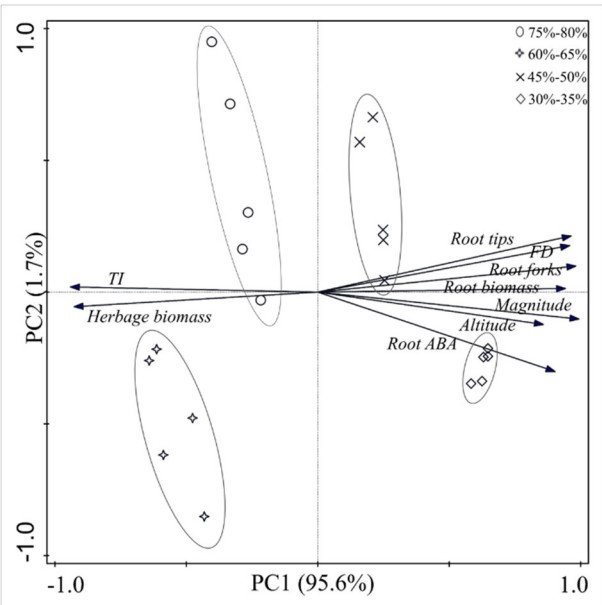

**Figure 8.** Principal component analysis of alfalfa root system characteristics (with water stress as the main effect) under soil water content treatments (75–80%, 60–65%, 45–50%, and 30–35% of water-holding capacity). PC1 accounted for 95.6% of the overall variance and PC2 accounted for 1.7% of the overall variance.

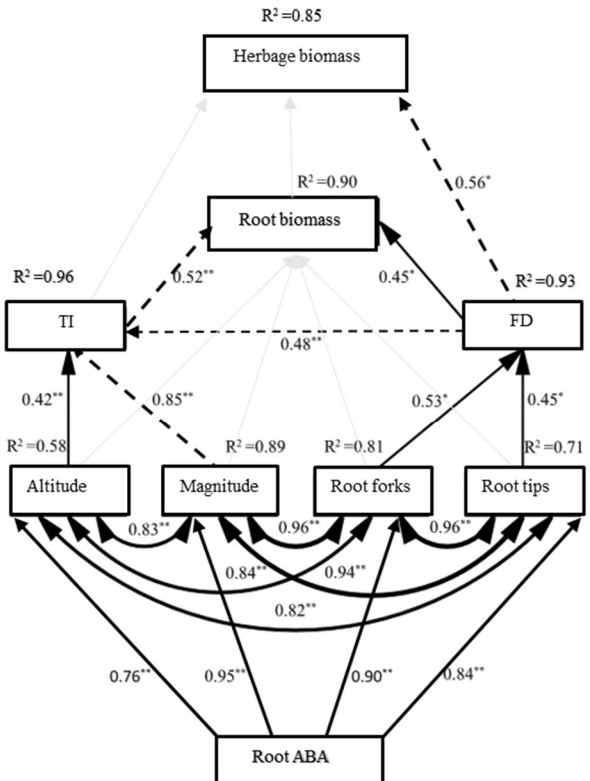

$X^2 = 8.501$ df = 8 $X^2$/df = 1.063 $P = 0.386$ RMSEA = 0.057 GFI = 0.901

**Figure 9.** Structural equation model (SEM) based on the effect of alfalfa root ABA content on altitude, magnitude, number of forks, number of root tips, topology index (TI), fractal dimension (FD), root biomass, and herbage biomass under various soil water content treatments ($\chi^2$ = 8.501, df = 8, $\chi^2$/df = 1.063, $P$ = 0.386, RMSEA = 0.057, GFI = 0.901). Continuous and dashed arrows indicate positive and negative relationships, respectively. Width of the arrows is proportional to the strength of

the path coefficients. As in other linear models, $R^2$ indicates the proportion of variance explained by the variables. The $R^2$ value is shown above every response variable in the model. Black and gray arrows indicate significant and non-significant correlations, respectively. The numbers beside arrows are the correlation coefficient between two indicators. * $p < 0.05$, ** $p < 0.01$.

## 4. Discussion

### 4.1. Effect of Water Stress

Water shortages are responsible for the greatest crop losses worldwide and are expected to worsen, heightening international interest in crop drought tolerance [6]. Drought occurs frequently in Northwest China and has a severe impact on crop yields. Alfalfa grows well in dryland regions and shows broad adaptability in the arid and semi-arid areas of Northwest China [3]. However, local alfalfa varieties show decreased production owing to reduced growth and survival caused by environmental stress. The findings of this study showed that the growing point density and herbage accumulation of alfalfa generally declined under more severe water deficit, which is supported by the study of Gramshaw et al. [43]. A soil water content of 60–65% of the water-holding capacity resulted in the highest alfalfa herbage production and residual shoot weight, and at this moisture level, AH and ZD had higher herbage accumulation and less root weight than GE in Experiment 1, which indicated that alfalfa can improve water-use efficiency by increasing herbage accumulation and reducing root weight under optimum soil water content. These different scenarios of drought have different impacts on pasture growth and development above and below ground. The same result has been also reported by Shan et al. [44]. Root growth, allocation, and distribution depend on plant growth strategies and the response to water deficit and distribution in the soil [6]. Plant root development is a crucial factor contributing to aboveground pasture production, persistence, and water-use efficiency in response to changes in environment conditions [45]. In our study, several morphological traits of root systems have been found to be associated with increased productivity under water deficit conditions, which indicated that tolerant plants can probably improve the health, growth, and functioning of the pasture, and interact with root traits to improve pasture yield [46,47]. We find that alfalfa showed increases in the number of root tips and root forks, and in root altitude and magnitude under severe water deficit in Experiment 2. This indicates that alfalfa has enhanced root development for acquisition of water from the soil under stressful water conditions [3]. In addition, AH and ZD had more herbage biomass and less root biomass under 60–65% of the water-holding capacity than severe water stress conditions, which indicated that the relation between the size of the aboveground part and roots is of vital importance for the plant water balance. The ability of plants to adjust root development according to the distribution of available soil water can potentially increase plant productivity under water stress. A similar conclusion was also reported in [48].

Some researchers [49] have developed theoretical approaches to evaluate topology, which indicate that a root system tends to dichotomous branching when the plant is in high-nutrient soil and the TI is close to 0.5; the branching pattern tends to herringbone branching when resources are scarce and the TI is close to 1 [50]. The main purpose of root topology measurement is to explore how root systems branching pattern changes in response to habitat conditions [51]. In our study, the topological analysis was used to quantify the root system architecture of alfalfa under different water deficit conditions. Under 60–65% of the water-holding capacity, AH and ZD had higher TI than GE in Experiment 2, which indicated that the root system of AH and ZD tend to exhibit a herringbone branching pattern with poor root branches under optimum soil moisture content (e.g., 60–65% of the water-holding capacity in this study). In contrast, alfalfa root had richer root branches under severe water stress, which indicated that root distribution range can be expanded by increasing the dichotomous branching pattern of secondary branches to enhance the absorption and utilization of nutrients. Day et al. [52] also found that tolerant plants can expand their root distribution range to adapt to severe stress conditions. Fractal

geometry has been widely used to evaluate root architecture and root distribution in soil. Fractal analysis can be a potentially useful tool for qualitatively describing root system morphological structures and accurately showing differences in plant complex root development [3,53]. The characterization of the fractal dimension (FD) is helpful for better understanding the function and growth strategy of plant roots when the soil water supply is insufficient [3,54]. FDs illustrate the differences in root system structures under various stress conditions, and can show the expanded volume of plant roots in soil [55]. The two-dimensional FD values obtained in this study were greatest when the soil moisture treatment was 30–35% of the water-holding capacity, demonstrating that alfalfa root systems had more branches and a larger volume range in order to uptake nutrients and water in response to severe water deficit. The root system of alfalfa under optimum soil water is a herringbone branch with small FD in this study. Li et al. [3] and Zhou et al. [56] also showed that root fractal characteristics and topological structure were correlated.

### 4.2. Abscisic Acid in Responses to Water Stress

Plant hormones are important secondary signaling molecules that mediate responses to many environmental stimuli, and some work has shown that several of these hormones promote growth primarily by acting in specific tissue layers [57] and regulate root meristem size [58,59]. The water stress-associated hormone abscisic acid (ABA) is a known inhibitor of lateral root development and has been shown to act at the postemergence stage. ABA is considered predominantly to be a root-derived regulator of plant growth in response to water stress [60] and is an important signal involved in root-to-shoot communication [29]. In our study, we examined that alfalfa roots contained higher ABA contents under 30–35% of the water-holding capacity. In contrast, under more than 60–65% of water-holding capacity treatments, the roots contained lower ABA content, which indicates that the production of ABA is induced under severe water stress and the production is necessary in response to water stress and drought tolerance. Similar results on the change in root ABA content were obtained in a previous study [61].

### 4.3. Correlation among Root ABA, Root System Architecture, and Plant Biomass

Abscisic acid is an important phytohormone that regulates plant growth, development, and stress responses. Extensive studies have revealed that ABA functions in a wide range of processes, including embryo and seed development, seed and shoot dormancy, root development, stress responses, and drought tolerance [62,63]. Interestingly, we examined that root ABA content was positively correlated with altitude, magnitude, number of root tips, and number of root forks of alfalfa root system under various water stress, which suggests that lateral root development was probably promoted by ABA in the roots in response to water deficit, which is in line with the findings of Smet et al. [63]. In addition, ABA regulates auxin homeostasis in the root tips to promote root hair elongation [64]. Root system architecture is well-known to be a major determinant of root functions in the acquisition of soil resources, such as nutrients and water [65], and thus greatly affects crop growth and yield [14,40]. The acquisition efficiency of a root system for water and nutrients is strongly affected by the spatial distribution of roots in the soil. The study of root system architecture has mainly focused on root geometry and topological parameters [3]. Fractal geometry is widely applied to assess root system architecture and the distribution of roots in soil [66]. In the present study, FD was positively correlated with root biomass and negatively correlated with herbage biomass, which indicate that FD was probably an indicator of the branching characteristics of the entire root system affecting plant biomass in response to water stress conditions. FD is correlated with root topology [42], and TI is used to reveal differences in branching patterns of root systems [3,67]. FD was negatively correlated with TI and assisted in understanding the spatial root forks and branching strategy of plant root systems when faced with water deficit. Thus, ABA in the roots indirectly regulate plant root spatial distribution as well as plant root biomass and herbage biomass. ABA is also believed to improve plant stress resistance. However,



studies of ABA have focused on the aboveground parts of plants and little research has been performed to investigate the belowground parts [68–70]. Future research needs to examine how root ABA can help plants to develop a larger root system to resist abiotic stresses and differences in drought avoidance should be evaluated for alfalfa cultivars that differ in root system architecture. In addition, the role of root-sourced ABA and root system architecture should be evaluated in field studies for practical implementations.

*4.4. Practical Implications and Further Research*

While the present findings have practical implications, it is important to note that the experiment was conducted under greenhouse conditions. The responses of alfalfa change when other environment factors are introduced in an arid or semi-arid area, therefore field validation of the results is essential. A crucial finding in this study is that severe water deficit had a negative effect on plant growth variables, including herbage accumulation, growing point density, residual shoot weight, and root weight for three alfalfa cultivars. This implies that continuous water deficit leads to rapid stand decline and affects regrowth ability and persistence of alfalfa. Alfalfa density under continuous water deficit treatments declines most rapidly [71]. The present study also revealed that AH had higher herbage accumulation and growing point density than that of GE and ZD under water stress treatments, which indicates the importance of genetic resources to management and environmental stress. At the morphological level, the shoots and roots are most affected and are critical components of plant adaptation to drought. Plants generally limit the number and area of leaves in response to water stress to reduce the water budget at the cost of yield loss [72]. Given that roots are the only source of acquiring water from the soil, the growth, density, proliferation, and size of the root system are crucial for plant in responses to water stress [73]. However, plant breeders generally shy away from selection for root traits because they show low heritability and their expression may vary with soil type and moisture content [74]. Plant breeders generally assume that direct selection for yield indirectly selects varieties with the optimum root system to produce the highest yields [75]. The highest herbage biomass and lowest root biomass were attained when the soil water content was optimal for specific cultivar in this study. A plant may change its root morphological traits to obtain higher amounts of water for plant growth [76]. Another important finding in the present study was that a strong root system of all cultivars for acquiring soil water was developed prior to the plant being suffered from severe water deficit; however, AH and ZD differed from GE in their ability to utilize water under water stress. It is possible that targeted selection for specific root architecture traits may enhance yields in dryland cropping regions. Further research is necessary in this regard, particularly studies under field conditions.

**5. Conclusions**

A total of 60–65% of the water-holding capacity resulted in rich alfalfa herbage accumulation, growing point density, and residual shoot weight. At the level of water stress, alfalfa root less root tips, root forks, altitude, and magnitude than severe water stress. Alfalfa roots tended to exhibit a herringbone branching pattern when the soil water content was 60–65% of the water-holding capacity. The cultivars ZD and AH had higher drought resistances than GE due to differences in root topological and fractal analysis. Abscisic acid in the roots was indirectly and directly involved in root development and plant growth. These findings help to understand alfalfa root system growth in response to water stresses and have important practical implications for managing and screening alfalfa cultivars, and field validation under different severities of water deficit and the mechanism by which root-sourced ABA regulates root development in response to environmental stress need further research.

**Supplementary Materials:** The following supporting information can be downloaded at: https://www.mdpi.com/article/10.3390/agronomy12081882/s1, Table S1: Mean herbage accumulation (HA; g DM pot-1) of alfalfa cultivars (AH—Aohan; GE—Golden Empress; ZD—Zhaodong) under various soil water content treatments (75–80%, 60–65%, 45–50%, and 30–35% of water-holding capacity); Table S2: Mean growing point density (GD; /pot) of alfalfa cultivars (AH—Aohan; GE—Golden Empress; ZD—Zhaodong) under various soil water content treatments (75–80%, 60–65%, 45–50%, and 30–35% of water-holding capacity); Table S3: Mean residual shoot biomass (SB, g DM pot-1) and root biomass (RB, g DM pot-1) of alfalfa cultivars (AH—Aohan; GE—Golden Empress; ZD—Zhaodong) under various soil water content treatments (75–80%, 60–65%, 45–50%, and 30–35% of water-holding capacity).

**Author Contributions:** Conceptualization and methodology, S.L. and G.Y.; writing, S.L.; review and editing, S.L., J.S. and Z.N.; investigation, S.L. and X.L. All authors have read and agreed to the published version of the manuscript.

**Funding:** This research was funded by the China Agriculture Research System (grant no. CARS-34) and the National Natural Science Foundation of China (project no. 31772671) and Doctoral Scientific Research Startup of Qingdao Agricultural University.

**Institutional Review Board Statement:** Not applicable.

**Informed Consent Statement:** Not applicable.

**Data Availability Statement:** Not applicable.

**Acknowledgments:** We thank Liqiang Wan, Lihong Miao and Wei Tang for their advice and technical assistance in conducting these studies. Zhensong Li, Xiaoxin Zhou, and Qian Gao were involved in some laboratory work for the study.

**Conflicts of Interest:** The authors declare no conflict of interest.

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
