# Peer review of "The Physiological Role of Abscisic Acid in Regulating Root System Architecture of Alfalfa in Its Adaptation to Water Deficit"

_agronomy, doi:10.3390/agronomy12081882_

Round 1

Reviewer 1 Report

This manuscript titled as "Effect of abscisic acid regulated root system architecture on alfaalfa adaptation in response to water deficit" is relevant to the journal, well written, and goes into details of various root morphological parameters that are influenced by drought conditions. Study is focused on three alfaalfa cultivars and 4 water stress treatments. Why these specific 4 water stress treatments were selected is not explained? The role of phytohormone ABA under water stress conditions was studied in detailed, however not enough literature related to ABA is provided in the introduction section. Another suggestion would be to add either actual photos of roots or root sketches highlighting various features such as topological index, altitude, magnitude, internal links, external links etc.

Author Response

pls see the attachment.

Reviewer 2 Report

The manuscript was well written. I made some minor changes. The study itself is quite interesting and was well done. Since both experiments were conducted in a greenhouse, it would be helpful for the authors to mention something in the limitation of this study in their discussion section. A well-controlled field experiment would have carried much more weight. Are you planning on such an experiment in the field? It may be nice to know.     

Author Response

pls see the attachment.
